# Flow Cytometric Methods for the Detection of Intracellular Signaling Proteins and Transcription Factors Reveal Heterogeneity in Differentiating Human B Cell Subsets

**DOI:** 10.3390/cells9122633

**Published:** 2020-12-08

**Authors:** Casper Marsman, Tineke Jorritsma, Anja ten Brinke, S. Marieke van Ham

**Affiliations:** 1Department of Immunopathology, Sanquin Research and Landsteiner Laboratory, Amsterdam UMC, University of Amsterdam, 1066 CX Amsterdam, The Netherlands; c.marsman@sanquin.nl (C.M.); t.jorritsma@sanquin.nl (T.J.); a.tenbrinke@sanquin.nl (A.t.B.); 2Swammerdam Institute for Life Sciences, University of Amsterdam, 1098 XH Amsterdam, The Netherlands

**Keywords:** differentiation, germinal center, antibody-secreting cells, phosphorylated STATs, NF-κB1

## Abstract

The flow cytometric detection of intracellular (IC) signaling proteins and transcription factors (TFs) will help to elucidate the regulation of B cell survival, proliferation and differentiation. However, the simultaneous detection of signaling proteins or TFs with membrane markers (MMs) can be challenging, as the required fixation and permeabilization procedures can affect the functionality of conjugated antibodies. Here, a phosphoflow method is presented for the detection of activated NF-κB p65 and phosphorylated STAT1, STAT3, STAT5 and STAT6, together with the B cell differentiation MMs CD19, CD27 and CD38. Additionally, a TF-flow method is presented that allows the detection of the B cell TFs PAX5, c-MYC, BCL6 and AID and antibody-secreting cell (ASC) TFs BLIMP1 and XBP-1s, together with MMs. Applying these methods on in vitro-induced human B cell differentiation cultures showed significantly different steady-state levels, and responses to stimulation, of phosphorylated signaling proteins in CD27-expressing B cell and ASC populations. The TF-flow protocol and Uniform Manifold Approximation and Projection (UMAP) analysis revealed heterogeneity in TF expression within stimulated CD27- or CD38-expressing B cell subsets. The methods presented here allow for the sensitive analysis of STAT, NF-κB p65 signaling and TFs, together with B cell differentiation MMs, at single-cell resolution. This will aid the further investigation of B cell responses in both health and disease.

## 1. Introduction

One of the cornerstones of the adaptive immune system is the ability of B cells to respond to pathogens and differentiate into antibody-secreting cells. The generation of antibody-secreting cells (ASCs) is mainly regulated by follicular T helper cells [1,2] (T_FH_) within germinal centers [3] (GCs) and is crucial for the development of high-affinity antibodies [4,5]. B cells are activated upon antigen recognition by the B cell receptor (BCR). Subsequently, the antigen is internalized and presented on MHC-II in order to form a cognate interaction with the T_FH_ cell [6]. This interaction provides multiple stimuli indispensable for GC B cell proliferation and differentiation, such as membrane-bound CD40L, which activates CD40 on the B cell [7,8], and the T_FH_ hallmark cytokine IL-21, which binds to the IL-21 receptor [9,10,11]. Furthermore, T_FH_ cells are also able to produce IL-4 [12,13], which contributes to class-switching and proper GC formation in combination with IL-21 [14,15,16]. When CD40L binds CD40 on the B cell surface, it induces the activation of the NF-κB p65 (canonical, heterodimerizes with NF-κB1) and NF-κB p52 (non-canonical, also known as NF-κB2) pathways [17,18,19]. The secreted cytokines IL-4 and IL-21 bind to their respective receptors and induce JAK–STAT pathways. IL-4 induces the phosphorylation of STAT6 [20,21,22], whereas IL-21 mainly induces the phosphorylation of STAT1, STAT3 and, to a lesser extent, STAT5 [22,23]. After the phosphorylation events, these signaling proteins translocate to the nucleus, where they can regulate transcription [24,25]. The induction of both the NF-κB and STAT pathways is essential, as mutations in involved genes are known to cause primary immunodeficiencies [10,26]. 

The regulation of transcription factors (TFs) in B cell and GC responses is highly complex and dynamic [27,28,29]. B cell identity is maintained by a high expression of PAX5 [30]. After stimulation by T cells, the GC is formed, and BCL6 and AID expression is upregulated in B cells [31]. Within the GC, c-MYC is crucial for the maintenance of the GC by regulating the division capacity and cycling of GC B cells [32,33,34]. Differentiation into ASCs requires the downregulation of PAX5, BCL6, AID and c-MYC and upregulation of BLIMP1 and XBP-1 [33,35]. Additionally, XBP-1 transcripts are alternatively spliced into XBP-1s transcripts for the efficient functioning of the unfolded-protein response [33,36]. As the GC response and differentiation process is highly dynamic, a range of TFs have to be simultaneously measured with currently known membrane markers (MMs) to allow the further investigation of heterogeneity in these responses and to further investigate the effect of stimulatory signals on the expression of these TFs. 

The regulation of TF expression through the intracellular signal transduction pathways activated by T_FH_-cell-provided CD40L, IL-21 and IL-4 is essential for the GC response and B cell differentiation. The ability to monitor both this signaling and transcriptional regulation is necessary to enable fundamental research on wanted B cell responses against pathogens and unwanted B cell responses, as observed in autoimmune diseases [37,38,39] or during allo-antibody formation after blood transfusion [40]. Additionally, transcriptional dysregulation and overactivated signaling pathways can lead to B cell malignancies [41,42]. Finally, it can aid in the development and characterization of therapeutic inhibitors for a range of diseases [43].

Recently, it was elegantly shown that the detection of BCR signaling by flow cytometry has a clear advantage over the detection of the BCR downstream signaling molecules by Western blotting [44]. The authors clearly show that phosphoflow analysis allows for a precise detection of BCR signaling at a single-cell resolution in a range of human and mouse B cell subsets. The next step for studying or targeting B cell differentiation in health and disease is combining the detection of the NF-κB and p-STAT signaling pathways activated by T_FH_ signals with MM analysis, to elucidate the crosstalk between BCR, CD40 and cytokine receptors. The detection of both MM and IC phospho-proteins has been optimized for analysis in murine lymphoid cells [45]. However, an optimized method for the simultaneous analysis of both MM and signaling in specific human B cell subsets undergoing differentiation is still highly desired. Furthermore, the detection of crucial TFs with MM expression during the T-dependent stimulation of B cells will also allow the further elucidation of human B cell differentiation. 

Here, we present two methods, one that allows the monitoring of signaling via STAT molecules and NF-κB and one that allows for the detection of TFs dynamically expressed throughout in vitro T-cell-dependent B cell differentiation. Both methods allow for sample processing in a 96-well format and allow for analysis at the level of individual B cells in distinct B cell subpopulations, revealing a previously unappreciated heterogeneity in stimulated B cell subpopulations.

## 2. Materials and Methods

### 2.1. Cell Lines

NIH3T3 fibroblasts expressing human CD40L (3T3-CD40L^+^) [46] were cultured in IMDM (Lonza, Basel 4002, Switzerland) containing 10% FCS (Serana, 14641 Pessin, Germany), 100 U/mL penicillin (Invitrogen, through Thermo Fisher, 2665 NN Bleiswijk, The Netherlands), 100 μg/mL streptomycin (Invitrogen), 2 mM l-glutamine (Invitrogen), 50 μM β-mercaptoethanol (Sigma Aldrich, 3330 AA Zwijndrecht, The Netherlands) and 500 μg/mL G418 (Life Technologies, through Thermo Fisher).

### 2.2. Isolation of Peripheral Blood Mononucleated Cells and B Cells from Human Healthy Donors 

Buffy coats of healthy human donors were obtained from Sanquin Blood Supply. All the healthy donors provided written informed consent in accordance with the protocol of the local institutional review board, the Medical Ethics Committee of Sanquin Blood Supply, and the study conformed to the principles of the Declaration of Helsinki. Peripheral blood mononucleated cells (PBMCs) were isolated from buffy coats using a Lymphoprep (Axis-Shield PoC AS, Dundee DD2 1XA, Scotland) density gradient. Afterwards, CD19^+^ B cells were isolated using magnetic Dynabeads (Invitrogen) and DETACHaBEAD (Invitrogen) according to the manufacturer’s instructions.

### 2.3. In Vitro B Cell Stimulation Cultures

3T3-CD40L^+^ were harvested, irradiated with 30 Gy and seeded in B cell medium (RPMI 1640 (Gibco, through Thermo Fisher) without phenol red containing 5% FCS, 100 U/mL penicillin, 100 μg/mL streptomycin, 2 mM L-glutamine, 50 µM β-mercaptoethanol and 20 µg/mL human apo-transferrin (Sigma Aldrich; depleted for human IgG with protein G sepharose (GE Healthcare, 3871 MV Hoevelaken, The Netherlands)) on 96-well flat-bottom plates (Nunc through Thermo Fisher) to allow adherence overnight. 3T3-CD40L^+^ were seeded at 10,000 cells per well. The next day, CD19^+^ B cells were thawed from cryo-storage and washed with B cell medium. B cells were rested at 37 °C for 1 h before counting. Then, 5000 or 10,000 CD19^+^ B cells were co-cultured with the irradiated 3T3-CD40L^+^ fibroblasts in the presence of F(ab’)_2_ fragment Goat Anti-Human IgA/G/M (5 µg/mL; Jackson Immunoresearch, Ely CB7 4EZ, UK), IL-4 (25 ng/mL; Cellgenix, 79107 Freiburg, Germany) and IL-21 (50 ng/mL; Peprotech, London W6 8LL, UK) for up to six days. The B cell suspension, 3T3-CD40L^+^ plates and stimulation mix were all heated to 37 °C before mixing the components together. After adding the B cells to the wells, the plate was centrifuged for 1 min at 400× *g* to force all the B cells onto the 3T3-CD40L^+^ layer.

### 2.4. Phosphoflow Protocol

#### 2.4.1. Flow Cytometry Antibodies

The antibodies used here were first titrated and validated. This was done by using either the manufacturers’ advised positive controls or by using a known strong stimulus found in literature [47,48]. During the validation and titration, the samples were compared to unstimulated and unstained controls. As the conditions and flow cytometer settings differ per lab, it is advised that these dilutions are taken as guidelines and that these are validated within each individual lab (Table 1).

#### 2.4.2. Harvesting, Fixation and Permeabilization

At the indicated timepoints, the cells in the wells were resuspended with a multichannel pipette and transferred to 96-well V-bottom plates (Nunc). Multiple culture wells were pooled, up to a full 96-well plate per replicate. After harvest, the cells were kept on ice or at 4 °C at all times. The cells were washed with 150 µL of ice-cold PBS/0.1% bovine serum albumin (BSA; Sigma Aldrich), centrifuged at 600× *g* for 2 min and pooled. Samples were stained in a 25 µL staining mix with 1:1000 LIVE/DEAD Fixable Near-IR Dead cell stain kit (Invitrogen) and anti-CD19 and CD38 antibodies (Table 1) diluted in ice-cold PBS/0.1% BSA, for 15 min on ice. The samples were washed once with 150 µL of ice-cold PBS/0.1% BSA, centrifuged at 600× *g* for 2 min and fixed with 37 °C 4% paraformaldehyde (PFA; Sigma) for 10 min at 37 °C. After fixation, the samples were centrifuged at 600× *g* for 2 min, washed once with 150 µL of ice-cold PBS/0.1% BSA and permeabilized with 90% methanol from a −20 °C freezer. The samples were incubated for at least 30 min or stored at −20 °C till the day of FACS analysis.

#### 2.4.3. Intracellular Staining and FACS Analysis

After permeabilization, samples were centrifuged at 600× *g* for 2 min, followed by two consecutive washes with 150 µL of ice-cold PBS/0.1% BSA. The samples were then stained in 25 µL of staining mix containing anti-CD27, anti-NF-κB p65, anti-p-STAT1, anti-p-STAT3, anti-p-STAT5 and anti-p-STAT6 (Table 1) diluted in PBS/0.1% BSA. The samples were incubated for 30 min on a plate shaker at room temperature. The samples were washed twice with 150 µL of PBS/0.1%BSA. Finally, the samples were resuspended in a volume of 150 µL, of which 100 µL was measured on a flow cytometer. The flow cytometer was calibrated by compensating for all conjugates using UltraComp eBeads compensation beads (Invitrogen). All the measurements were performed on a BD FACSymphony machine and analyzed using the FlowJo Software v10.6.2 (Treestar). 

### 2.5. Real-Time Semiquantitative RT-PCR

Different B cell subsets (as indicated) were sorted on FACSAriaIII. After sorting, RT-PCR was performed as described before [49]. Briefly, cells were lysed in peqGOLD Trifast (PeQlab, 91052 Erlangen, Germany), and GlycoBlue (Ambion, 61440 Oberursel, Germany) was added as a carrier. Total RNA was extracted according to the manufacturer’s instructions. First-strand cDNA was reverse transcribed using random primers (Invitrogen) and SuperScript™ II Reverse Transcriptase (Invitrogen) according to the manufacturer’s instructions. The primers were developed to span exon–intron junctions and then validated. Gene expression levels were measured in duplicate reactions for each sample in StepOnePlus (Applied Biosystems, through Thermo Fisher) using the SYBR Green method with Power SYBR Green (Applied Biosystems, through Thermo Fisher). The primer sets used were as follows:c-MYC: F: 5′-TACAACACCCGAGCAAGGAC-3′      R: 5′GAGGCTGCTGGTTTTCCACT-3′
Published previously [23]:PA5: F: 5′-ACGCTGACAGGGATGGTG-3′,    R: 5′-CCTCCAGGAGTCGTTGTACG-3′BCL6: F: 5′-GAGCTCTGTTGATTCTTAGAACTGG-3′   R: 5′-GCCTTGCTTCACAGTCCAA-3′BLIMP1: F: 5′-AACGTGTGGGTACGACCTTG-3′        R: 5′-ATTTTCATGGTCCCCTTGGT-3′XBP-1: F: 5′-CCGCAGCACTCAGACTACG-3′,    R: 5′-TGCCCAACAGGATATCAGACT-3′AICDA: F: 5′-GACTTTGGTTATCTTCGCAATAAGA-3′       R: 5′AGGTCCCAGTCCGAGATGTA-3′
Expression was normalized to the internal control of 18S rRNA [49]:18S-rRNA: F: 5′-CGGCTACCACATCCAAGGAA-3′         R: 5′-GCTGGAATTACCGCGGCT-3′

### 2.6. TF-Flow Protocol

Cells were harvested, pooled and pelleted before washing twice with 10 mL of PBS/0.1% BSA. The samples were counted, and 1 × 10^6^ cells were added per well to a 96-well V-bottom plate. The samples were centrifuged at 600× *g* for 2 min and stained with 25 µL of staining mix with 1:1000 LIVE/DEAD Fixable Near-IR Dead cell stain kit and anti-CD19, anti-CD27 and anti-CD38 antibodies (Table 1) and incubated for 15 min in the fridge. The samples were washed once with 150 µL of ice-cold PBS/0.1% and centrifuged at 600× *g* for 2 min. The samples were then fixated with 100 µL of Foxp3 fixation buffer (eBioscience, through Thermo Fisher) for 30 min in the fridge. Next, 150 µL of Foxp3 permeabilization buffer (eBioscience) was added, and the samples were centrifuged at 600× *g* for 2 min. After removing the supernatant, the samples were stained with 25 µL of staining mix containing anti-PAX5 and anti-BCL6, and anti-c-MYC, anti-BLIMP1, anti-XBP-1s or anti-AID (Table 1) diluted in Foxp3 permeabilization buffer and incubated for 30 min in the fridge. The samples were washed with 150 µL of Foxp3 permeabilization buffer. Finally, the samples were resuspended in a volume of 140 µL, of which 90 µL was measured on a flow cytometer. The flow cytometer was calibrated by compensating for all conjugates using UltraComp eBeads compensation beads (Invitrogen). All the measurements were performed on a BD FACSymphony machine and analyzed using the FlowJo Software v10.6.2 (Treestar). 

### 2.7. Multimarker Analysis Using UMAP

Live CD19^+^ B cells were gated on, and duplicates from three donors were randomly down-sampled to 10,000 events and subsequently concatenated into a single 60,000 event FCS file using the DownSample plugin in FlowJo v10.6.2. Next, the concatenated sample was analyzed using the Uniform Manifold Approximation and Projection (UMAP) plugin v3.1 in FlowJo v10.6.2. UMAP is a machine-learning algorithm used for dimensionality reduction to visualize high-parameter datasets in a two-dimensional space. The UMAP plugin settings were as follows: distance function, Euclidean; number of neighbors, 30; minimal distance, 0.5; and number of components, 2. The UMAP dot plot generated can be manipulated as a standard dot plot and allows for multiple parameter and heatmap overlays in the FlowJo layout editor. The DownSample and UMAP FlowJo plugins can be found on the FlowJo Exchange website.

### 2.8. Statistical Analysis

Statistical analysis was performed using GraphPad Prism (version 8; GraphPad Software). The phosphoflow data were analyzed with mixed-effects analysis with Tukey’s multiple-comparison test. TF mRNA and flow data were analyzed with repeated-measures ANOVA with Tukey’s multiple-comparison test. Results were considered significant at *p* < 0.05. Significance was depicted as * *p* < 0.05, ** *p* < 0.01 and **** *p* < 0.0001.

## 3. Results

### 3.1. In Vitro Stimulation of B Cells

An in vitro stimulation system that mimics T-cell-dependent B cell stimulation (Figure 1A) was used to functionally assess the NF-κB and STAT signaling and TF profiles during B cell differentiation within specific B cell and ASC subpopulations. In this culture system, F(ab)_2_ fragments targeting IgM, IgG and IgA were used to induce BCR signaling. Irradiated human-CD40L-expressing 3T3-fibroblasts were used to mimic the T_FH_-mediated CD40 co-stimulation essential for B cell differentiation. Additionally, recombinant IL-21 and IL-4 cytokines were added to induce IL-21R and IL-4R signaling. B cells were cultured for up to 6 days, and B cell differentiation was assessed based on the relative expression of CD27 and CD38 (Figure 1B; full gating strategy in Appendix A). The relative expression of CD27 and CD38 (Figure 1C,D) demonstrated that significant B cell differentiation into the CD27^+^CD38^+^ ASC population started after 96 h and increased over time. 

### 3.2. Establishment of Phosphoflow Assay Shows Differences in Phospho-Signaling Profiles within B Cell Subpopulations

To be able to investigate the dynamics of STAT and NF-κB signaling during B cell differentiation, a phosphoflow protocol and panel were designed to allow the simultaneous detection of MM differentiation together with intracellular signaling via p-STAT and activated NF-κB p65. The validation and titration of the conjugated antibodies used here are listed in Table 1. Several antibodies directed against MMs for B cell differentiation and their conjugated fluorophores were tested for compatibility with the PFA-based fixation and methanol-based permeabilization protocol. Violet fluorophores such as BV510 and V450, together with the FITC and APC fluorophores, survived permeabilization with only a slight loss in fluorescence (Appendix A). Additionally, the anti-CD27 clone L128 was found to stain cells efficiently even after fixation and permeabilization. Even though some stains were significantly less after permeabilization or showed a significantly lower mean fluorescent intensity (MFI), the CD19, CD27 and CD38 populations were still easily distinguished. These antibodies and fluorophore properties combined allowed for an extensive panel for staining both MM and IC signaling proteins using the protocol presented here.

For the detection of p-STAT and activated NF-κB signaling, B cells were stimulated as above. CD19^+^ and CD27/CD38 subpopulations were gated (Figure 2A) and analyzed for the expression of p-STAT1, p-STAT3, p-STAT5, p-STAT6 and activated NF-κB p65. The analysis of p-STAT1 per CD27/CD38 subpopulation showed a clear induction of STAT1 signaling in all the populations over time, with the CD27^+^CD38^+^ ASC population reaching its maximum pSTAT1 value at around 48 h. By contrast, pSTAT1 peaked at 72 h in the three B cell populations, with higher and more sustained p-STAT1 levels in the CD27^−^CD38^+^ population (Figure 2B).

p-STAT3 levels increased over time, with a peak at 72 h of stimulation (Figure 2C). There were significant differences between subpopulations, with a tendency for the higher phosphorylation of STAT3 in the CD27^−^CD38^−^ population.

p-STAT6 levels in the B cell subpopulations were significantly different at the start of the culture up until 6 h of stimulation (Figure 2D). p-STAT6 increased in all the subpopulations until 48 h after stimulation. After this peak, the CD27^−^CD38^−^ and CD27^−^CD38^+^ populations remained higher and showed more sustained phosphorylation of STAT6 compared to the other populations. 

NF-κB p65 was activated as early as after 30 min of stimulation, followed by a second and higher peak of activation at 48–72 h of stimulation. (Figure 2E). To accommodate the differences in fluorescence intensity between donors (Appendix A), the fold change of NF-κB p65 was plotted to compare the dynamics of NF-κB p65 activation over time between the subpopulations. After 72 h of stimulation, NF-κB p65 levels declined in all subpopulations. There was a tendency for higher and more sustained NF-κB p65 activation in the CD27^−^CD38^−^ and CD27^−^CD38^+^ populations. The significant differences between the pSTAT and NF-κB p65 levels in the CD27/CD38 subpopulations could not be attributed to differences in cell size (Appendix A). Altogether, analysis showed that this panel can be used to study the dynamics of the combined pSTAT and NF-κB p65 signaling cascades during B-cell-to-ASC differentiation.

p-STAT5 analysis in the B cell subpopulations showed a gradual and transient increase in STAT5 phosphorylation over time, with a prominent early induction in the CD27^+^CD38^+^ ASC-population as early as after 6 h of stimulation (Figure 3A). Interestingly, a comparison of the four donors showed varying p-STAT5 profiles in the CD27^+^CD38^+^ ASC populations, resulting in a high deviation of GeoMFI (Appendix A). To further investigate this, the -CD19^+^ p-STAT5 high population (FI > 1000) was gated on in two donors at multiple timepoints (Figure 3B–D). Even though these two donors showed different profiles of p-STAT5 within subpopulations at a GeoMFI level, the p-STAT5^high^ cells found in both donors at several timepoints had similar CD27/CD38 expression profiles (Figure 3B–D, right). This demonstrates that these p-STAT5^high^ cells were indeed induced in both donors at early timepoints in the CD27^+^CD38^+^ ASC-population, but that these could be overlooked when averaging the values within subpopulations. Altogether, the data clearly show that the simultaneous analysis of pSTAT5 and CD27 and CD38 differentiation markers in single cells allows for a more in-depth interrogation of B cell signaling across specific subpopulations, compared to using CD19 alone. 

### 3.3. Sorted Naïve and Memory B Cells Show Significant Differences in Response to Varying CD40L, IL-21 and/or IL-4 Stimuli

Next, naïve (CD19^+^ CD27^−^ IgD^+^) and memory B cells (CD19^+^ CD27^+^) were sorted to further dissect intracellular signaling in naïve and memory B cells upon the induction of B cell differentiation with various stimuli (Figure 4A). 

Clear p-STAT1 induction was observed through activation with CD40L in both naïve and memory B cells. The added presence of IL-21 further increased p-STAT1 levels in memory B cells (Figure 4B). Although the induction of p-STAT1 coincided with increased cell size, cell size alone did not fully account for the p-STAT1 induction in the memory B cells (Appendix A). 

The induction of p-STAT3 was the result of CD40L stimulation and was amplified when IL-21 was also present in both naïve and memory B cells, irrespective of the presence of IL-4 (Figure 4B,C). 

Although BCR ligation yielded p-STAT5 signaling, CD40L was the strongest inducer of p-STAT5 signaling, with naïve B cells being superior in p-STAT5 induction to memory B cells (Figure 4B,C). 

In line with previous data [20,21,22], p-STAT6 was induced by IL-4, as early as after 30 min (Figure 4C). This induction was amplified by CD40L. In addition, the induction of p-STAT6 was significantly higher in naïve B cells compared to memory B cells. Activated NF-κB p65 was clearly induced when CD40L was present, as early as after 30 min of stimulation (Figure 4B,C). In addition, activated NF-κB p65 was significantly higher in naïve B cells compared to memory B cells after 72 h of stimulation. p-STAT3, 5 and 6 were induced by specific stimuli irrespective of cell size (Appendix A). The higher induction of activated NF-κB p65 in naïve compared to memory B cells also indicates that this is induction that cannot fully be attributed to increased cell size. Altogether, these data show that the method presented here can be utilized to analyze the effects of crosstalk between BCR signaling and T_FH_-derived CD40 and cytokine co-stimulation regarding STAT and NF-κB signaling in both naïve and memory B cells.

### 3.4. TF-Flow Assay Allows for a High-Resolution Analysis of TFs within Subpopulations

Next, a flow cytometric assay to detect known B cell transcription factors (Figure 5A) within B cell subpopulations was set up. To validate the TF-flow analyses, the TF protein expression levels measured by flow were compared to the mRNA expression levels measured by semiquantitative PCR. First, after a 6-day stimulation, the B cell subpopulations were sorted based on CD27 and CD38 expression (Figure 5B). Subsequently, the mRNA expression of the transcription factors PAX5, c-MYC, BCL6, BLIMP1, XBP-1 and AICDA was measured by semiquantitative PCR (Figure 5C). As expected, the mRNA expression of PAX5, c-MYC and AICDA was significantly downregulated in the CD27^+^CD38^+^ population compared to other populations. Additionally, the expression of BLIMP1 and XBP-1 was increased. Altogether, this indicates that the CD27^+^CD38^+^ B cells transitioned from a B cell mRNA expression profile to an ASC mRNA expression profile as the B cell signature gene PAX5 was downregulated and ASC-related genes BLIMP1 and XBP-1 were upregulated (Figure 5A). Next, the same experiment was performed but, then, the transcription factors were stained within the different B cell subpopulations (Figure 5C,D). As the reagents in the Foxp3 staining kit greatly reduce cell size (making it impossible to gate on a lymphocyte population), the cells were first gated on live cells before doublet exclusion and gating for the CD19 and CD27/CD38 subpopulations. The protein expression profiles of all the transcription factors within the subpopulations are highly comparable to the mRNA data. The expression of TFs was not dependent on cell size (Appendix A). Furthermore, there was a clear downregulation of CD19 in the CD27^+^CD38^+^ ASC population that coincided with a downregulation of PAX5 and an upregulation of BLIMP1 (Appendix A). The clear advantage of these TF analyses by flow cytometry is that they allow for the detection of protein instead of mRNA. Another advantage is the added depth of the analyses, as demonstrated by the observed bimodal expression of PAX5 and BLIMP1 within the CD27^+^CD38^−^ and CD27^−^CD38^+^ populations (Figure 5E, right), which could not be detected in the mRNA expression analysis. 

### 3.5. UMAP Analysis Unravels B Cell Subpopulation Heterogeneity

To further investigate the heterogenous expression of transcription factors within the B cell subpopulations on a single-cell level, a UMAP analysis was performed on the flow cytometric data (Figure 6). Cells from either the CD27^−^CD38^−^ population or the CD27^+^CD38^+^ population clustered together, as seen in the UMAP overlay of the B cell subpopulations (Figure 6A) and the heatmap expression plots (Figure 6B). Neither the CD27^+^CD38^−^ population nor the CD27^−^CD38^+^ population formed a uniform cluster. Notably, and in line with the mutually exclusive expression patterns of PAX5 (denoting the B cell signature) and BLIMP1 (denoting the ASC signature), the heatmaps of PAX5 and BLIMP1 clearly show a mirrored pattern, where PAX5 expression is reduced the BLIMP1 expression is increased. Strikingly, the UMAP analyses bring to light a small cluster of high-BCL6-expressing cells located on the border of the CD27^+^CD38^+^ cluster (Figure 6B, lower-right). As this population is not noted when plotting only the GeoMFI of the CD27/CD38 subsets, these data show the clear added benefit of applying UMAP analyses on data generated with this TF-flow method. As the expression of PAX5 and BLIMP1 was bimodal in the CD27^+^CD38^−^ and CD27^−^CD38 populations (Figure 5E), this heterogeneity was further elucidated by comparing the heatmap expression of CD27, PAX5 and BLIMP1 (Figure 6C). The data show that the expression of CD27 coincided with BLIMP1 expression (and therefore negatively coincided with PAX5 expression). Similar patterns are seen for CD38 in the CD27^−^CD38^+^ population. Together, these heatmaps explain the bimodal expression seen in Figure 4E. Altogether, these data show that the TF-flow protocol can be used to efficiently stain TFs within B cell subpopulations. This technique allows for a relatively fast analysis of TF expression at a single-cell resolution, a major advantage over mRNA-expression analysis, where populations have to be sorted. In addition, this method and UMAP analysis allow for the uncovering of small populations that have unique TF expression profiles.

## 4. Discussion

Here, we present a protocol for the detection and monitoring of signaling via phosphorylated STAT and activated NF-κB p65 within several B cell subpopulations. Furthermore, a second method is presented here that allows for the detection of TFs within B cell subpopulations. Both flow cytometry-based methods provide a clear advantage over other techniques such as Western blotting or the detection of mRNA. First, cells do not have to be purified prior to analysis, and MMs can be used to distinguish differentiated B cell subpopulations. Furthermore, these methods allow for analysis at a single-cell level for more in-depth analyses of cell signaling and TF expression in specific cellular subpopulations. In addition, these methods are optimized for sample processing in a 96-well format, allowing for a high-throughput analysis. Finally, one major advantage of this methanol-based phosphoflow method is that samples can be stored in the freezer for prolonged periods of time [50]. This allowed for the harvesting at multiple timepoints over the course of a 6-day culture and staining the intracellular phospho-proteins of multiple samples simultaneously. One drawback of these methods is the lack of commercially available conjugated antibodies with a wide range of fluorophores. Not all fluorophores or antibodies maintain reactivity under the phosphoflow protocol, as also noted by others [44,50,51]. Additionally, antibodies directed against nuclear TFs are currently only available for a small selection of fluorophores. This makes it difficult to combine several TF-targeting antibodies into a single panel. When setting up the transcription flow within one’s own lab, it is advised to reserve the brightest fluorophores, such as PE-Cy7 and AF647, for antibodies targeting the nuclear TFs. This ensures the highest resolution of the detection of these, sometimes little expressed, TFs. Fortunately, the availability of antibodies and fluorophores is constantly expanding, and this will most likely allow for a more extensive panel in the near future. 

Using these phosphoflow and TF-flow methods, several observations were made. First, B cell subsets showed significantly different steady-state levels of phosphorylated STAT1 and STAT6. Mainly, the CD27^+^CD38^−^ subset, conventionally called memory B cells, and the CD27^+^CD38^+^ ASC population had significantly higher levels of phosphorylated STAT1 and STAT6. Furthermore, all the measured intracellular signaling proteins were more rapidly induced and also declined earlier in these subsets compared to the CD27^−^CD38^−^ and CD27^−^CD38^+^ populations. Interestingly, when naïve and memory B cells are compared, p-STAT6 induction was much higher in naïve B cells compared to memory B cells. In both subsets, p-STAT6 was induced by IL-4, as shown previously [20,21,22], and this induction was amplified if CD40L was present. In addition, p-STAT5 and activated NF-κB p65 levels were significantly higher in naïve B cells compared to memory B cells under specific stimuli. These data confirm previous findings that NF-kB p65 induction is higher in naïve compared to memory B cells [51]. Furthermore, in a different in vitro stimulation system, it was shown that p-STAT5 was more rapidly induced in naïve B cells compared to memory B cells [52]. However, in that paper, IL-21 was used as a stimulus. The data presented here show that IL-4 in fact induced and maintained higher levels of p-STAT5 compared to IL-21, as there was no dip in p-STAT5 at the 24 h timepoint when IL-4 was present. Furthermore, it was previously shown that p-STAT1 and p-STAT3 induction was higher in naïve B cells and was mediated mainly by IL-21 [23,52]. This contradicts the data presented here showing that p-STAT1 and p-STAT3 levels were higher in memory B cells compared to naïve B cells after stimulation. In addition, p-STAT1 was already induced by CD40L alone and further increased by IL-21 in memory B cells. Similarly, the data presented here show that CD40L already induced p-STAT3 in both memory and naïve B cells, and that this induction was amplified by IL-21. The main differences between the previously mentioned in vitro system [52] and the one utilized here are that the authors had to sort the splenic B cell subsets instead of circulating B cells and that these cells were pre-stimulated with anti-Ig/CD40L before adding cytokines and measuring the phosphorylation of signaling proteins. Another observation is the rapid induction of p-STAT5 in the CD27^+^CD38^+^ ASC population in the first hours of stimulation. As these high-p-STAT5 CD27^+^CD38^+^ cells were almost all gone after 24 h, it is likely that this induction was specific to ex vivo circulating plasmablasts present in the culture. These cells were, however, not sustained within this system. To the best of our knowledge, this induction has not been shown before. The regulation of p-STAT5 in B cells remains largely unknown. It has been shown that p-STAT5 can induce but also inhibit BCL6 expression [53,54,55]. As BCL6 is tightly linked to the survival and proliferation of B cells [56], this induction of p-STAT5 in the CD27^+^CD38^+^ ASC population could be linked to the survival of these cells after in vitro stimulation. Additionally, cells with high levels of p-STAT5 found throughout the culture may have a different capacity to survive and proliferate compared to the other cells. Combining the phosphoflow method here with cell trace proliferation dyes could help to further elucidate this. To ensure that the differences in the signaling profiles in the stimulated B cell subsets were not due to differences in cell size, the FSC-A parameter was investigated as an indication of size. It was shown that, indeed, after prolonged culture and stimulation, the FSC-A increased. However, this increase did not correlate with the increases in p-STAT and NF-κB levels, showing that the induction was specific to the stimuli provided. As the increased cell size could lead to increased total-STAT levels, these should be investigated and compared to p-STAT induction. This was not conducted here, as the flow panel did not allow the addition of antibodies directed against all total-STAT proteins.

The TF-flow method revealed a previously unappreciated heterogeneity of PAX5 and BLIMP1 expression in the CD27^+^CD38^−^ and CD27^−^CD38^+^ cells. Furthermore, a high-BCL6-expressing population was found that could not have been detected by measuring mRNA in bulk sorted populations. As a high expression of BCL6 is a hallmark of GC B cells, this high-BCL6 population could be in vitro-induced GC B cells. Together, these data show that the single-cell resolution gained with this flow cytometric method will allow for the further in-depth identification of distinct B cell subsets and elucidation of specific intracellular signaling and TF expression over time. 

The methods presented here allow for a sensitive and efficient analysis of signaling proteins and TFs, together with MMs. These methods allow for analysis at a single-cell resolution that could aid the immunomonitoring of signaling and transcriptional regulation in healthy and harmful B cell responses, such as rheumatoid arthritis, lupus, vasculitis and allo-antibody formation [37,38,39,40]. Additionally, these techniques could aid in the development and characterization of therapeutic inhibitors [43]. Furthermore, as different B cell subsets have been known and shown here to have varying and dynamic regulation of signaling [52,57], the methods presented here will help fundamental research to further investigate this.

## Figures and Tables

**Figure 1 cells-09-02633-f001:**
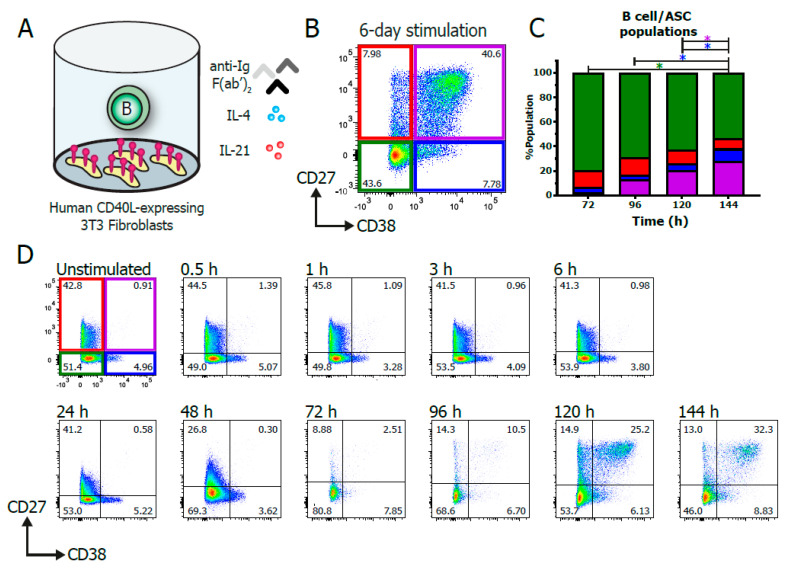
Efficient in vitro generation of human B cell and antibody-secreting cell subpopulations. Total human B cells (*n* = 3) were stimulated in vitro to generate multiple B cell and antibody-secreting cell (ASC) subpopulations. (**A**) Schematic overview of the culture system used to induce B cell differentiation. A total of 5000 CD19^+^ human B cells were stimulated with a human-CD40L-expressing 3T3 feeder layer; an anti-Ig F(ab)_2_ mix (5 μg/mL) targeting IgM, IgG and IgA; and recombinant IL-4 (25 ng/mL) and IL-21 (50 ng/mL) cytokines. (**B**) Representative FACS plot after 6 days of culture based on expression of CD27 and CD38. (**C**) Quantification of the relative percentages of CD27 and CD38 subpopulations in the total CD19^+^ B cell population between 3 and 6 days of culture. *n* = 4; *p* values were calculated by 2-way ANOVA with Tukey’s multiple-comparison test; * *p* < 0.05. (**D**) Representative FACS plots of a time-course experiment of B cell differentiation dynamics of one B cell donor, as measured over the course of 6 days of culture.

**Figure 2 cells-09-02633-f002:**
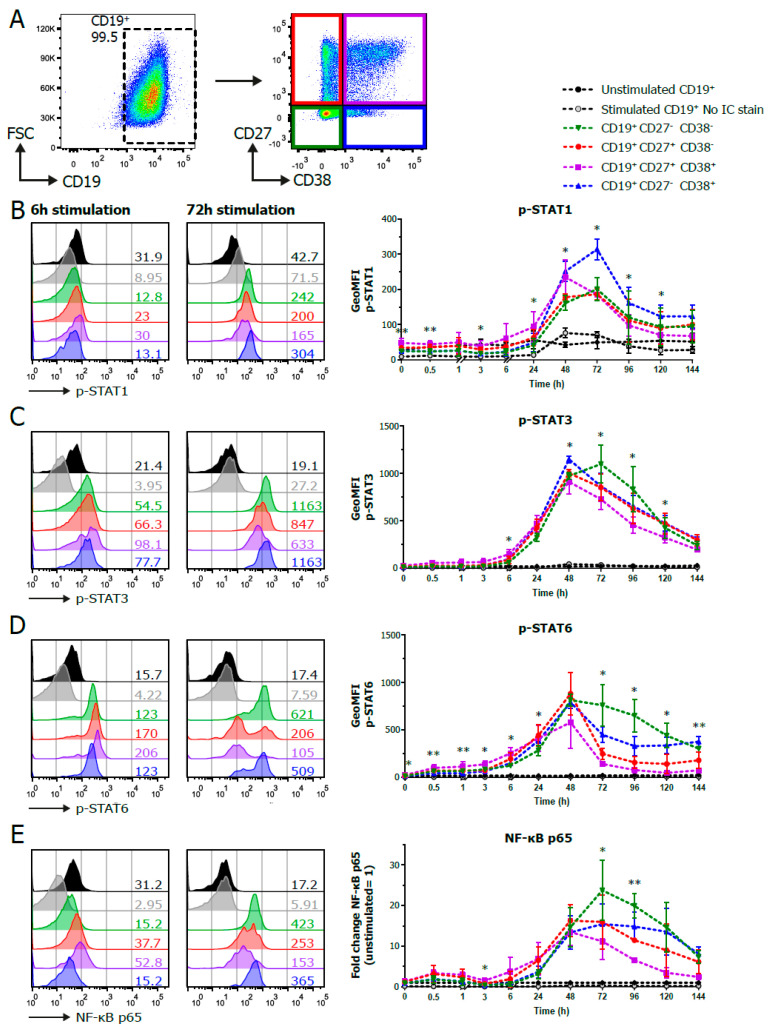
Phosphoflow analysis of stimulated human B cells show differences between CD27/CD38 subpopulations. Human B cells (*n* = 3–4) were stimulated with a human-CD40L-expressing 3T3 feeder layer, an anti-Ig F(ab)_2_ mix (5 μg/mL) targeting IgM/IgG/IgA and recombinant IL-4 (25 ng/mL) and IL-21 (50 ng/mL) cytokines, and multiple signaling proteins were analyzed by phosphoflow analysis over the course of a 6-day culture. (**A**) Representative FACS plots show the gating strategy for CD19^+^ and CD27/CD38 subpopulations by phosphoflow analysis after 96 h stimulation. (**B**–**E**) Representative histogram overlays of p-STAT1 (**B**), p-STAT3 (**C**), p-STAT6 (**D**) and NF-κB p65 (**E**) staining in unstimulated and stimulated CD19^+^ and CD27/CD38 subpopulations after 6 and 72 h stimulation (left), and the quantification of the geometric MFI (GeoMFI) within the different subpopulations over the course of 6 days of culture (right). Values depicted next to histograms represent the corresponding GeoMFI. For NF-κB p65 (E), fold change was calculated by normalizing to the expression in unstimulated CD19^+^ cells (set at value of 1). *n* = 3–4; *p* values were calculated using a mixed-effect analysis with Tukey’s multiple-comparison test; * *p* < 0.05 and ** *p* < 0.01. Significance * is depicted if there was a significant difference between the green, red, purple or blue CD27/CD38 subpopulations.

**Figure 3 cells-09-02633-f003:**
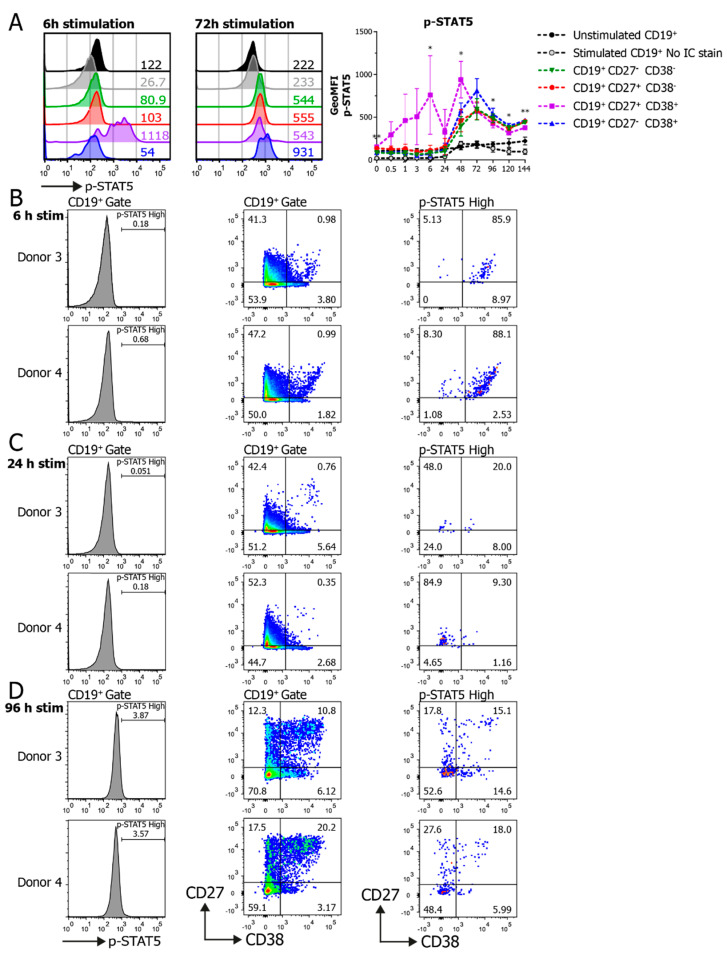
Phosphoflow analysis of STAT5 phosphorylation unveils heterogeneity over time in CD27/CD38 subpopulations. Human B cells (*n* = 3–4) were stimulated with a human-CD40L-expressing 3T3 feeder layer, an anti-Ig F(ab)_2_ mix (5 μg/mL) targeting IgM/IgG/IgA and recombinant IL-4 (25 ng/mL) and IL-21 (50 ng/mL) cytokines, and multiple signaling proteins were analyzed by phosphoflow analysis over the course of a 6-day culture. (**A**) Representative histogram overlays of p-STAT5 staining in CD19^+^ and CD27/CD38 subpopulations after 6 and 72 h stimulation (left), and the quantification of the GeoMFI of p-STAT5 within the different subpopulations over the course of 6 days of culture (right). Values depicted next to histograms represent the corresponding GeoMFI. *n* = 3–4; *p* values were calculated using a mixed-effect analysis with Tukey’s multiple-comparison test; * *p* < 0.05 and ** *p* < 0.01. Significance is shown if there was a significant difference between the green, red, purple or blue CD27/CD38 subpopulations. (**B**–**D**) Histograms of p-STAT5 expression in the CD19^+^ population in Donor 3 and Donor 4 (left), and the corresponding CD27/CD38 expression profiles in the CD19^+^ population (middle) and high-p-STAT5-expressing population (right) after 6 h (**B**), 24 h (**C**) or 96 h (**D**) of stimulation.

**Figure 4 cells-09-02633-f004:**
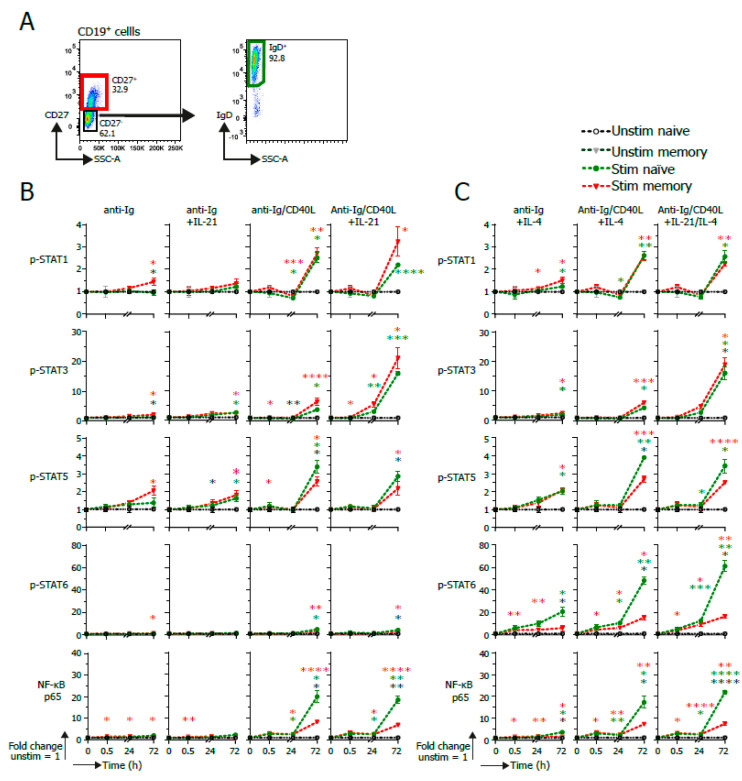
pSTAT and NF-κB signaling in naïve and memory B cells upon B cell activation via BCR, CD40 or IL4/Il-21. A total of 10,000 human naïve (CD19^+^ CD27^−^ IgD^+^) or memory (CD19^+^ CD27^+^) B cells (*n* = 3) were stimulated after sorting with an anti-Ig F(ab)_2_ mix (5 μg/mL) either together or not together with CD40L-expressing 3T3 cells and recombinant IL-4 (25 ng/mL) and/or IL-21 (50 ng/mL) cytokines. Multiple signaling proteins were analyzed by phosphoflow analysis up to 72 h of stimulation. (**A**) Representative FACS plots of the sorting strategy for purification of CD19^+^ CD27^+^ memory and CD19^+^ CD27^−^ IgD^+^ naïve B cells. (**B**) Quantification of GeoMFI of signaling proteins in stimulated sorted naïve or memory B cells after 30 min, 24 h or 72 h stimulation with varying IL-21 stimulations. Fold change was calculated normalizing expression to unstimulated condition. *p* values were calculated using a mixed-effect analysis with Tukey’s multiple-comparison test. * *p* < 0.05, ** *p* < 0.01, *** *p* < 0.001 and **** *p* < 0.0001. Red * shows significance of stimulated memory compared to unstimulated, green * shows significance of stimulated naïve compared to unstimulated, and black * shows significance of stimulated naïve vs. stimulated memory. (**C**) Quantification of GeoMFI of signaling proteins in stimulated sorted naïve or memory B cells after 30 min, 24 h or 72 h stimulation with varying IL-4 stimulations. Fold change was calculated normalizing the expression to the respective unstimulated condition. *p* values were calculated by using a mixed-effect analysis with Tukey’s multiple comparison test. * *p* < 0.05, ** *p* < 0.01, *** *p* < 0.001and **** *p* < 0.0001. Red * shows significance of stimulated memory compared to unstimulated, green * shows significance of stimulated naïve compared to unstimulated, and black * shows significance of stimulated naïve vs. stimulated memory.

**Figure 5 cells-09-02633-f005:**
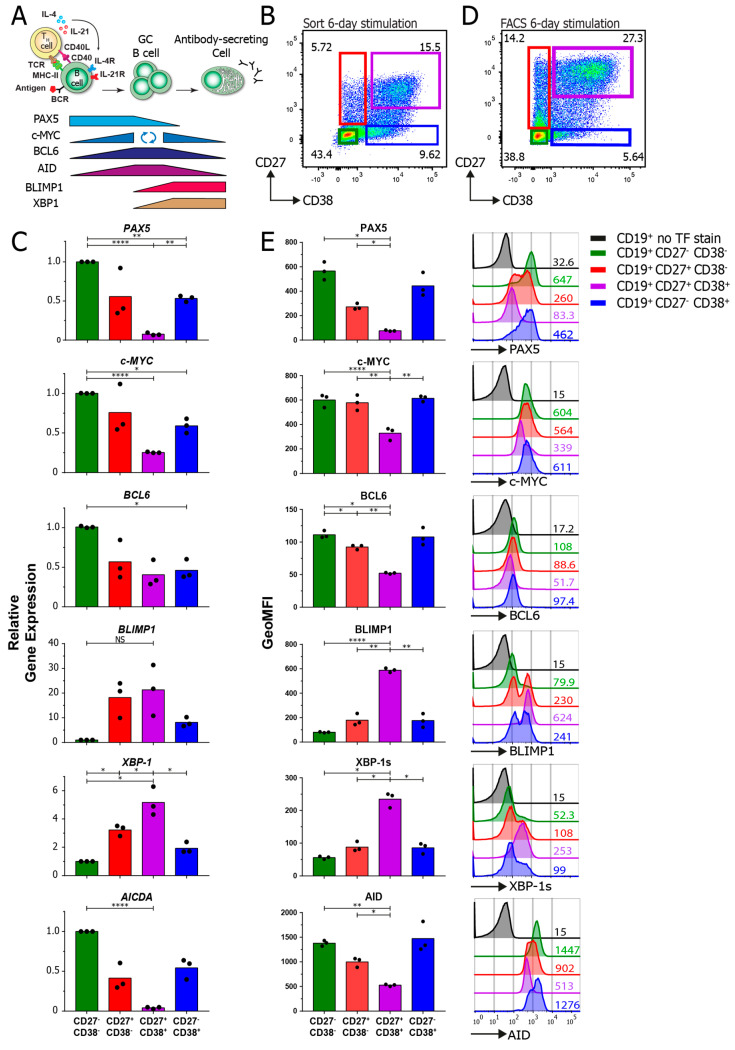
Transcription factor analysis in stimulated human B cells by flow cytometry. Human B cells (*n* = 3) were stimulated with an anti-Ig F(ab)2 mix (5 μg/mL) and recombinant IL-4 (25 ng/mL) and IL-21 (50 ng/mL) cytokines for 6 days and analyzed for the mRNA and protein expression of multiple transcription factors. (**A**) Schematic representation of the expression of B cell-, GC cell- and ASC-defining transcription factors after T cell-dependent B cell stimulation. (**B**) Representative FACS plot of the sorting strategy for the CD27/CD38 subpopulations for analysis by semiquantitative RT-PCR. A more stringent gating strategy was used here to prevent the contamination of subpopulations during sorting. (**C**) Quantification of relative gene expression of *PAX5*, *c-MYC*, *BCL6*, *BLIMP1*, *XBP-1* and *AICDA* in different CD27/CD38 subpopulations as measured by semiquantitative RT-PCR. All results were normalized to the internal control 18S rRNA. Expression was calculated relative to the CD27^−^CD38^−^ subpopulation (set at value of 1). Each dot represents an independent donor, and mean values are represented as bars (*n* = 3). (**D**) Representative FACS plot of the CD27/CD38 subpopulations that were stained for transcription factors and analyzed by FACS. (**E**) Quantification of the GeoMFI of PAX5, c-MYC, BCL6, BLIMP1, XBP-1s (active spliced isoform) and AID stained for in different CD27/CD38 subpopulations (left) and corresponding histogram overlays (right). Values depicted next to histograms represent the corresponding GeoMFI. Each dot represents an independent donor, and mean values are represented as bars (*n* = 3). *p* values were calculated using Tukey’s multiple-comparison test, * *p* < 0.05, ** *p* < 0.01 and **** *p* < 0.0001.

**Figure 6 cells-09-02633-f006:**
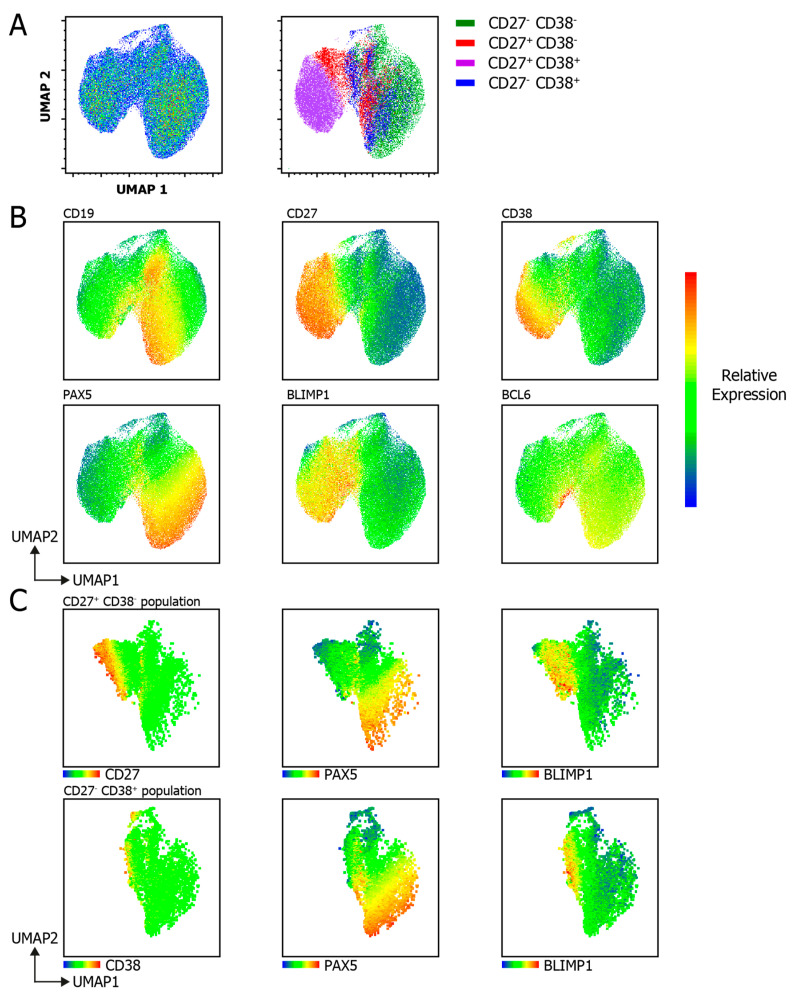
Uniform Manifold Approximation and Projection (UMAP) analysis of combined membrane marker and TF marker analyses upon B cell differentiation. UMAP clustering analysis on B cells stained for CD19, CD27, CD38, PAX5, BLIMP1 and BCL6 after 6-day culture with a human-CD40L-expressing 3T3 feeder layer, an anti-Ig F(ab)_2_ mix (5 μg/mL) targeting IgM/IgG/IgA and recombinant IL-4 (25 ng/mL) and IL-21 (50 ng/mL) cytokines. (**A**) UMAP 2D scatter plot of 60,000 living, single CD19^+^ B cells from 3 donors (left), and the different CD27/CD38 subpopulations overlaid (right). UMAP settings were as follows: distance function, Euclidean; number of neighbors, 30; minimal distance, 0.5; and number of components, 2. (**B**) Heatmap of the relative protein expression of CD19, CD27, CD38, PAX5, BLIMP1 and BCL6 included in the UMAP analysis overlaid on the UMAP 2D scatter plot. (**C**) Heatmap of the relative protein expression of PAX5 and BLIMP1 within the isolated CD27^+^CD38^−^ population (top) and CD27^−^CD38^+^ population (bottom) overlaid on the UMAP 2D scatter plot.

**Table 1 cells-09-02633-t001:** Antibodies used for phospho-specific and transcription factor flow cytometry.

Antibody Target	Conjugate	Clone	Manufacturer	Dilutions *	Cat. No.
**Membrane Markers**					
**CD19**	APC	SJ25C1	BD	1:400	345,791
	BV510	SJ25C1	BD	1:100	562,947
**CD27**	APC	L128	BD	1:50	337,169
	PE	L128	BD	1:50	340,425
	BUV395	L128	BD	1:100	563,815
	BUV737	L128	BD	1:100	612,829
**CD38**	V450	HB7	BD	1:100	646,851
	FITC	T16	Beckman Coulter	1:50	A07778
**Phosphoflow**					
**pSTAT1**	Percp-Cy5.5	4a pY701	BD	1:5	560,113
**pSTAT3**	PE	4/P-STAT3	BD	1:5	612,569
**pSTAT5**	Pacific Blue	47/Stat5 (pY694)	BD	1:5	560,311
**pSTAT6**	AF647	18/P-Stat6	BD	1:5	612,601
**NF-κB p65**	PE-Cy7	K10-895.12.50	BD	1:25	560,335
**Transcription Factors**					
**PAX5**	PE	1H9	Biolegend	1:10.000	649,708
**c-MYC**	AF647	D84C12	CST	1:150	13871S
**BCL6**	PE-Cy7	7D1	Biolegend	1:400	358,512
**BLIMP1**	AF647	#646702	R&D	1:40	IC36081R-025
**XBP-1s**	AF647	Q3-695	BD	1:40	562,821
**AID**	AF647	EK2-5G9	BD	1:150	565,785

* Optimal antibody dilutions as defined for the method and staining procedure used in this paper. As the staining conditions and flow cytometer settings may differ per lab, it is advised that these dilutions are taken as guidelines and that these are validated within each individual lab. See Materials and Methods for the full staining procedure.

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
