# Peer review of "Flow Cytometric Methods for the Detection of Intracellular Signaling Proteins and Transcription Factors Reveal Heterogeneity in Differentiating Human B Cell Subsets"

_cells, 2020, doi:10.3390/cells9122633_

Round 1

Reviewer 1 Report

This manuscript details the use of phospho-flow cytometric techniques to examine phosphorylation profiles of several well-known signaling molecules, including transcription factors, in order to elucidate differences in signaling architecture between human B cell subsets.  From a technical perspective the experiments are well performed.  Proper care seems to have been taken to insure fundamental data reproducibility, and sufficient detail as to experimental techniques has been reported so as to allow other labs to reproduce the experiments.  Overall the basic finding of this manuscript is that there is heterogeneity in the phosphorylation of signaling molecules as B cells differentiate from circulating B cells to plasma cells, after cells are stimulated in vitro and then cultured.  Unfortunately, in spite of a superb experimental technique, the paper is still lacking.  The problem is that taken as a whole, the paper is essentially descriptive in nature.  Although the specific phospho-targets were selected by virtue of their previously know function in B cell signaling pathways, the experimental design gives little insight into the mechanism of B cell signaling, other than to say that there differences in signaling between the examined B cell subsets.  The authors have set the stage for a more in depth analysis of B cell signaling, but have not followed through.

Apart from fundamental issues of B cell signaling in this particular model system, the authors posit that their methods will now allow for analysis of B cell signaling in a wide variety of systems, including RA, lupus, vasculitis and allo-antibody formation.  The problem here is that the use of phospho-flow cytometry to study B cell signaling was first introduced about a decade ago by the Nolan lab.  Since that time there have been multiple reports of phopho-flow cytometry to study lymphocyte signaling.  Most recently, updated protocols for phospho-flow analysis of B cells have been reported by J. Pip et al. (reference 45 in this paper).  If the authors are now claiming that their methods are a new standard for phospho-flow approaches in the study of B cell signaling, it is absolutely essential that they document superior performance of their approach over preceding protocols in side by side comparisons.  This has not been done. 

Reviewer 2 Report

Authors have presented a beautiful study on analyzing transcription factors using flow cytometry. This is an excellent work that is presented well.

There are no major comments.

Minor comments.

  1. Introduction for this manuscript is a bit long and can be made shorter.
  2. If possible, it will be helpful to the reader to provide some of the TF flow staining data by cytospin/cytology staining. this is to understand the spatial location of these TF. Tonsil tissues or spleen if available could be used.

Reviewer 3 Report

In this study, the authors describe results for intracellular detection of phosphoprotein and transcription factors applying to an in vitro B cell differentiation model. Although this study contains several interesting aspects, it is written in a quite convoluted way that it does not facilitate the reading. 

The use of intracellular markers to define more accurately the different steps of the immune cell differentiation is widely used in some lineages (myeloid and T cells) but remains poorly exploited in B cells. The work here is an attempt to propose an additional strategy beyond classical membrane marker detection to classify and explore B-cell differentiation and the heterogeneity of antibody-secreting cells. 

 Unfortunately, most of the work presented is very fragmented, with many imprecisions and lacks appropriate interpretations or even conclusions. Most of the time titles of figures are overrated and are not supported by the results. 

To a large extent, one problem resides in the lack of purification of B cell subsets before the culture leading to an inevitable heterogeneity in results as shown in Figure 1. The ASC percentage fluctuates from 10 to 40 % depending on the donors. We can assume that this difference may be related to the proportion of naive and memory B cell subsets before the culture. It seems to me that a more definitive understanding of the processes at play in these cultures would require the purification of naive and memory B cell subsets to perform cultures with discrete B cell subsets rather than total B cells. 

In consequence, most of the conclusions are superficial. They do not add much to what is known currently. There is no attempt to rely on some interesting findings like the early increase of P-STAT5 with a specific mechanism.  

Major comments: 

The term extracellular (EC) is incorrect, I guess the authors meant membrane markers. Please correct.

-Authors should increase their number of experiments: 3 is quite limited, adding experiments of sorting B-cell subsets would increase the potential of interests.

Figure 1 

-Figure 1 D: Number of cells representing « cells expansion » is the result of the proliferation and apoptosis occurring during the in vitro process. Accurate cell proliferation measurement should be done to offer a better overview of the dynamic expansion of B cells in culture. Furthermore, measurement of B cells counts from unstimulated B cells after 6 days of culture is quite useless, it is expected that most of the cells (certainly all) will die after a few days.

Figure 1E is confused since the different subsets have a very different range of expression and don’t add to the story.

As validation of their model, authors could present results of antibodies production (Elispot, supernatant Elisa).

Please think to represent the B-cell populations independently or present only one population of interest (maybe CD27+CD38+?).

Figure 2

Figure 2 A, The representation of CD27/CD38 gating is already shown in figure 1 A. 

Since B cells are differentiating in culture, it is very surprising to don’t see the CD19 downregulation following the rise of plasmablasts.

Figure 2 B-E Graphs are not comprehensive, please present the significant results with clear conclusions.

Figure 3

The early upregulation of pSTAT5 is interesting. This increase is observed at 6 H but seems to disappear after 24H of stimulation. How the authors interpret these observations?

Is it possible that the early activation observed comes from plasmablasts already present in the B cells before the culture? Figure 1 C showed that the proportion of CD27+ CD38+ is the same at baseline and within the 6h first hours of culture and then decreases until 72H suggesting that cells are already present and don’t result from the culture differentiation.

Again, it appears crucial to separate B cell populations before the culture to ensure proper interpretation and conclusions.

Figure 4

An important effort to demonstrate that mRNA correlates with protein expression has been made and this makes sense although we can assume their dynamic of variation differs. 

Could authors add representative flow cytometry data of TF staining in the PB (CD27+ CD38+) and the non-activated B cells subset (CD27-, CD38-) with a CD19 expression by example? 

Figure 5

UMAP or tSNE are interesting representation when a lot of markers are involved and both of them excel at maximizing the resolution between populations which don’t stain brightly or discretely. However, it seems that the title of the figure is a little overrated. Authors apply the four already identified discrete subsets based on CD27 and CD38 expression on the UMAP plots : It is not the elucidation of heterogeneity within B cell subpopulations. 

Conclusions:

 Although, we could understand that the purpose of this study was to expose some results from an alternative technical method. We could expect that authors provide additional interpretations and new insights bringing by this method. 

I am convincing that the rational is good and that these methods would increase understanding of the B-cell differentiation pathway, but for now, the results present here are not convincing and deserve to be reworked.

Round 2

Reviewer 1 Report

Adequate modifications from the original submission have been made in the revised submission.